# Theoretical-Computational Modeling of Gas-State Thermodynamics in Flexible Molecular Systems: Ionic Liquids in the Gas Phase as a Case Study

**DOI:** 10.3390/molecules27227863

**Published:** 2022-11-14

**Authors:** Andrea Amadei, Andrea Ciccioli, Antonello Filippi, Caterina Fraschetti, Massimiliano Aschi

**Affiliations:** 1Dipartimento di Scienze e Tecnologie Chimiche, Università di Roma “Tor Vergata”, Via della Ricerca Scientifica 1, 00133 Roma, Italy; 2Dipartimento di Chimica, Università di Roma, “La Sapienza”, P.le A. Moro 5, 00185 Roma, Italy; 3Dipartimento di Chimica e Tecnologie del Farmaco, Università di Roma, “La Sapienza”, P.le A. Moro 5, 00185 Roma, Italy; 4Dipartimento di Scienze Fisiche e Chimiche, Università de l’Aquila, Via Vetoio (Coppito 2), 67010 l’Aquila, Italy

**Keywords:** ionic liquids, thermodynamics, molecular dynamics

## Abstract

A theoretical-computational procedure based on the quasi-Gaussian entropy (QGE) theory and molecular dynamics (MD) simulations is proposed for the calculation of thermodynamic properties for molecular and supra-molecular species in the gas phase. The peculiarity of the methodology reported in this study is its ability to construct an analytical model of all the most relevant thermodynamic properties, even within a wide temperature range, based on a practically automatic sampling of the entire conformational repertoire of highly flexible systems, thereby bypassing the need for an explicit search for all possible conformers/rotamers deemed relevant. In this respect, the reliability of the presented method mainly depends on the quality of the force field used in the MD simulations and on the ability to discriminate in a physically coherent way between semi-classical and quantum degrees of freedom. The method was tested on six model systems (n-butane, n-butane, n-octanol, octadecane, 1-butyl-3-methylimidazolium hexafluorophosphate and 1-butyl-3-methylimidazolium bis(trifluoromethylsulfonyl)imide ionic pairs), which, being experimentally characterized and already addressed by other theoretical-computational methods, were considered as particularly suitable to allow us to evaluate the method’s accuracy and efficiency, bringing out advantages and possible drawbacks. The results demonstrate that such a physically coherent yet relatively simple method can represent a further valid computational tool that is alternative and complementary to other extremely efficient computational methods, as it is particularly suited for addressing the thermodynamics of gaseous systems with a high conformational complexity over a large range of temperature.

## 1. Introduction

The accurate theoretical-computational modeling of thermodynamic properties such as the standard Gibbs free energy, enthalpy, heat capacity, and entropy for molecular or supramolecular species in the gas phase is a longstanding problem [1], which has received, in the last years, a great deal of attention [2,3,4,5], not only because of great fundamental interest, but also for its practical importance for better understanding and, sometimes, predicting the physical and chemical stability of compounds in a wide temperature and pressure range. The reliability of the modeling of these properties, addressed in the context of statistical mechanics, relies on two basic ingredients: (i) a good molecular Hamiltonian capable of describing in great detail the molecular or supramolecular system under investigation and, when necessary, (ii) the possibility of exhaustively sampling the associated configurational space. The first condition is nowadays achieved by a wide variety of theoretical-computational strategies at affordable computational costs, which also use quantum-chemical calculations. [6,7,8,9,10] The only limitation for an accurate outcome of these approaches for rigid gaseous species, which is their systematic use of the harmonic approximation [11], is now easily circumvented through several strategies proposed in the last years [12,13,14,15,16]. On the other hand, when flexible molecules or molecular clusters are concerned, the scenario becomes much more complicated and computationally problematic because of the large, in some cases prohibitively large, associated configurational space, which must be properly sampled. In this respect, different methods have been proposed and successfully applied, ranging from those including the torsional anharmonicity by means of the approximation of uncoupling the torsional energies, which are particularly suitable for low-sized molecular systems [17,18,19,20,21], to methods inspired by the ‘minima mining’ approach, which are potentially capable of dealing with large-scale systems [22,23,24]. In this context, the methods recently presented by Grimme and coworkers [5] or by Suarez and coworkers [25] are particularly efficient, as they allow the calculation of absolute entropies, heat capacities and reaction free energies, even in flexible molecular species. These latter methods are based, inter alia, on semi-classical molecular dynamics (MD) simulations, which, in the presence of an accurate empirical force field for the system of interest, can provide proper configurational space sampling, relevantly reducing the difficulty of arbitrarily searching for the different energy minima (i.e., the accessible conformations). Inspired by these latter approaches and as a part of the continuing interest of one of us [26] in the study of the thermodynamics of gaseous molecular species, we herein propose a theoretical-computational strategy, which, from a technical point of view, starts from MD semi-classical simulations and, in this perspective, can be considered as a further example of the ensemble of methodologies just described [5,25]. However, the present approach shows two specific peculiarities. From a practical point of view, our method allows us to automatically treat any molecular species with a large—even very large—conformational associated repertoire, i.e., bypassing any type of identification, extraction and ex post analysis (by means of, e.g., quantum chemical calculations) of the different conformers/rotamers sampled along a semi-classical simulation. Moreover, from a more genuine theoretical point of view, our method is based on the quasi-gaussian entropy (QGE) theory [27,28,29,30], in whose context the basic statistical mechanical relations are completely redefined in terms of distributions of the fluctuations of macroscopic properties, instead of the partition function. More specifically, in QGE and, in general, in statistical mechanics, it is possible to define a proper reference state such that the free energy difference between the actual condition and the reference state can be expressed in terms of the moment-generating function of the distribution of a specific macroscopic fluctuation. The modeling of such a moment-generating function allows one to describe the thermodynamics of the system of interest, avoiding the explicit calculation of the partition function. Therefore, in our theoretical-computational approach, we do not need to exhaustively search for all the local minima, which is typically the difficulty of the methods present in the literature, often making their application very difficult in complex molecular systems and/or in wide temperature ranges. Moreover, the use of MD simulation data coupled to the QGE theoretical approach allows us to include in our evaluation of the system thermodynamics all the anharmonic effects due to the conformational sampling outside the quasi-harmonic basins (especially relevant at high temperatures for complex molecular systems). The present study is organized as follows. In the first part, we report, in some detail, the theoretical framework underlying the proposed method; in the second part, the computational and technical details are outlined. In order to test the quality of the presented approach, in the final part of the study, we address six test cases that are experimentally characterized and already addressed by other computational methods, making them particularly suited for comparisons capable of bringing out advantages and possible drawbacks. In particular, we focused our attention on (i) n-butane (C${}_{4}$H${}_{10}$), which has been extensively investigated from an experimental [31,32] and computational [33] point of view; (ii) two supramolecular systems of great interest to one of us [34,35], namely, the ion pairs 1-butyl-3-methylimidazolium hexafluorophosphate (BmimPF${}_{6}$) and 1-butyl-3-methylimidazolium bis(trifluoromethylsulfonyl)imide (BmimNTf${}_{2}$), as reported in the Figure 1, which are particularly difficult to computationally address because of their relatively wide configurational accessible space [18,36,37]; (iii) n-butanol (C${}_{4}$H${}_{10}$O), which has been extensively investigated with computational tools [20,38] different from the one presented in this study and also experimentally characterized [39]; (iv) octadecane (C_18_H_38_), a particularly challenging flexible system successfully addressed in a recent theoretical study by Grimme and coworkers [5]; and (v) n-octane (C${}_{8}$H${}_{18}$), an experimentally [40,41] and computationally [5] well-studied hydrocarbon with a relatively high internal flexibility.

## 2. Theory

### 2.1. The Gamma State Model

Let us consider a homogeneous macroscopic (ideal) gas-state system, made of flexible molecules or molecular complexes (i.e., they possess semi-classical internal degrees of freedom) that we always consider chemically stable (in the following, we define such molecules or molecular complexes as system *molecules*). The ideal gas condition (no interactions among the *molecules*) allows us to obtain the system thermodynamics considering a single *molecule*, either in the canonical ensemble within a fixed volume corresponding to the system *molecular* volume or in the isobaric–isothermal ensemble with a fixed pressure identical to the system equilibrium pressure. In the isobaric–isothermal (N,p,T) ensemble, the Gibbs free energy of such a single *molecule* system (i.e., the chemical potential μ) is given by
(1)$$\mu \left(p,T\right)=-k_{B}T\ln\Delta \left(p,T\right)$$
where $k_{B}$ is the Boltzmann constant and *T* is the absolute temperature. The isobaric–isothermal partition function Δ(p,T) can be expressed as [30]
(2)$$\begin{array}{ccc} \Delta (p,T) & = & \sum_{V}Q\left(V,T\right)e^{-\beta pV} \end{array}$$
(3)$$\begin{array}{ccc} Q(V,T) & ≅ & \Theta Q_{vb}\left(T\right)\int_{V}e^{-\beta [U_{e}\left(q\right)+K\left(q,\pi \right)]}d\Gamma \end{array}$$
(4)$$\begin{array}{ccc} Q_{vb}\left(T\right) & = & \sum_{l}e^{-\beta E_{vb,l}} \end{array}$$
where *p* is the equilibrium pressure, $1/\beta =k_{B}T$. The summation in Equation (Equation 2) is over all the possible volumes *V* of the system (the difference between two consecutive volumes is virtually a differential). $Q_{vb}\left(T\right)$ is the *molecular* quantum vibrational partition function defined by the vibrational energies $E_{vb,l}$, and the subscript *V* of the integral sign means that integration is performed within the volume *V*. Moreover, $U_{e}\left(q\right)$ is the electronic ground state energy (the electronic excited states are disregarded, as they are virtually inaccessible except at extremely high temperatures), K(q,π) is the classical kinetic energy, Θ is a constant providing the quantum correction for the permutations of identical particles (possibly including the degeneration factor of the electronic ground state) and $d\Gamma =dqd\pi /h^{n}$ expresses the number of semi-classical quantum states within the phase space differential, with *h* being Planck’s constant, *n* being the number of the *q*-generalized semi-classical coordinates and π the corresponding conjugated momenta. Therefore, from Equations (Equation 1)–(Equation 3), we can write
(5)$$\beta \mu \left(p,T\right)≅-\ln Q_{vb}\left(T\right)-\ln\sum_{V}\Theta \int_{V}e^{-\beta [U_{e}\left(q\right)+K\left(q,\pi \right)+pV]}d\Gamma$$

In order to obtain the thermodynamics as a function of the temperature (i.e., along an isobar), we can use Equation (Equation 5) to express βμ as a function of the temperature variation $T_{0}\to T$ (i.e., corresponding to $\Delta \beta =\beta -\beta_{0}$)
(6)$$\begin{array}{ccc} \beta \mu \left(p,T\right)-\beta_{0}\mu \left(p,T_{0}\right) & ≅ & -\ln\frac{Q_{vb}\left(T\right)}{Q_{vb}\left(T_{0}\right)}-\ln\frac{\sum_{V}\int_{V}e^{-\beta [U_{e}\left(q\right)+K\left(q,\pi \right)+pV]}d\Gamma }{\sum_{V}\int_{V}e^{-\beta_{0}\left[U_{e}\left(q\right)+K\left(q,\pi \right)+pV\right]}d\Gamma }\\ =-\ln\frac{Q_{vb}\left(T\right)}{Q_{vb}\left(T_{0}\right)} & - & \ln\frac{\sum_{V}\int_{V}e^{-\beta_{0}\left[U_{e}\left(q\right)+K\left(q,\pi \right)+pV\right]}\;e^{-\Delta \beta [U_{e}\left(q\right)+K\left(q,\pi \right)+pV]}d\Gamma }{\sum_{V}\int_{V}e^{-\beta_{0}\left[U_{e}\left(q\right)+K\left(q,\pi \right)+pV\right]}d\Gamma }\\ & = & -\ln\frac{Q_{vb}\left(T\right)}{Q_{vb}\left(T_{0}\right)}-\ln{\langle e^{-\Delta \beta [U_{e}\left(q\right)+K\left(q,\pi \right)+pV]}\rangle }_{\beta_{0}} \end{array}$$
where ${\langle e^{-\Delta \beta [U_{e}\left(q\right)+K\left(q,\pi \right)+pV]}\rangle }_{\beta_{0}}$ is the moment-generating function of the single phase space position enthalpy (i.e., $U_{e}\left(q\right)+K\left(q,\pi \right)+pV$) and the $\beta_{0}$ subscript of the angle brackets means averaging within the $\beta_{0}$ ensemble. From Equation (Equation 6), defining the excess Gibbs free energy as
(7)$$\mu^{\prime }=-k_{B}T\ln\sum_{V}\Theta \int_{V}e^{-\beta [U_{e}\left(q\right)+K\left(q,\pi \right)+pV]}d\Gamma ≅\mu +k_{B}T\ln Q_{vb}$$
we readily obtain
(8)$$\begin{array}{ccc} \beta \mu^{\prime }\left(p,T\right)-\beta_{0}\mu^{\prime }\left(p,T_{0}\right) & = & -\ln\frac{\sum_{V}\int_{V}e^{-\beta [U_{e}\left(q\right)+K\left(q,\pi \right)+pV]}d\Gamma }{\sum_{V}\int_{V}e^{-\beta_{0}\left[U_{e}\left(q\right)+K\left(q,\pi \right)+pV\right]}d\Gamma }\\ & = & -\ln{\langle e^{-\Delta \beta [U_{e}\left(q\right)+K\left(q,\pi \right)+pV]}\rangle }_{\beta_{0}} \end{array}$$
clearly showing that the excess free energy change can be expressed by the moment-generating function (MGF) of the distribution of the single phase space position enthalpy, necessarily diverging for β→0 (see Equation (Equation 8)). In fluid state systems, a typically accurate and physically fully acceptable distribution is the Gamma distribution [27,28,42], providing for the molecular excess free energy, entropy, enthalpy and (isobaric) heat capacity of the diverging Gamma state expressions [29,30,43]
(9)$$\begin{array}{ccc} \mu^{\prime }\left(p,T\right) & = & h_{0}^{\prime }-T_{0}c_{p0}^{\prime }+T\left(c_{p0}^{\prime }-s_{0}^{\prime }\right)+Tc_{p0}^{\prime }\ln\frac{T_{0}}{T}\\ & = & \mu_{0}^{\prime }+\left(T-T_{0}\right)\left(c_{p0}^{\prime }-s_{0}^{\prime }\right)+Tc_{p0}^{\prime }\ln\frac{T_{0}}{T} \end{array}$$
(10)$$\begin{array}{ccc} s^{\prime }\left(p,T\right) & = & s_{0}^{\prime }+c_{p0}^{\prime }\ln\frac{T}{T_{0}} \end{array}$$
(11)$$\begin{array}{ccc} h^{\prime }\left(p,T\right) & = & h_{0}^{\prime }+c_{p0}^{\prime }\left(T-T_{0}\right) \end{array}$$
(12)$$\begin{array}{ccc} c_{p}^{\prime }\left(p,T\right) & = & c_{p0}^{\prime } \end{array}$$
with
(13)$$\begin{array}{ccc} \mu_{0}^{\prime } & = & \mu^{\prime }\left(p,T_{0}\right) \end{array}$$
(14)$$\begin{array}{ccc} s_{0}^{\prime } & = & s^{\prime }\left(p,T_{0}\right) \end{array}$$
(15)$$\begin{array}{ccc} h_{0}^{\prime } & = & h^{\prime }\left(p,T_{0}\right) \end{array}$$
(16)$$\begin{array}{ccc} c_{p,0}^{\prime } & = & c_{p}^{\prime }\left(p,T_{0}\right) \end{array}$$

Equations (Equation 9)–(Equation 12) provide the excess thermodynamics along an isobar (i.e., as a function of the temperature at a fixed pressure), according to the diverging Gamma state model; once at a given arbitrary reference temperature $T_{0}$, the corresponding excess entropy, enthalpy and heat capacity (i.e., $s_{0}^{\prime }$, $h_{0}^{\prime }$ and $c_{p0}^{\prime }$) are known. It is worth noting that the linear temperature dependence obtained for the excess enthalpy (i.e., due to the constant excess heat capacity along the isobar) can be used as the diagnostic criterion for validating the diverging Gamma state as a proper model for the isobar thermodynamics. Once an explicit expression of the molecular vibrational partition function $Q_{vb}$ is available from Equations (Equation 7) and (Equation 9), we can readily obtain the complete molecular Gibbs free energy (i.e., the chemical potential) along the isobar
(17)$$\begin{array}{ccc} \mu (p,T) & ≅ & \mu^{\prime }\left(p,T\right)-k_{B}T\ln Q_{vb}\left(T\right)\\ & ≅ & \mu_{0}^{\prime }+\left(T-T_{0}\right)\left(c_{p0}^{\prime }-s_{0}^{\prime }\right)+Tc_{p0}^{\prime }\ln\frac{T_{0}}{T}+\sum_{j}\frac{h\nu_{j}}{2}\\ & + & k_{B}T\sum_{j}\ln\left(1-e^{-\beta h\nu_{j}}\right) \end{array}$$
and thus, via its temperature derivatives, the corresponding molecular full enthalpy, entropy and (isobaric) heat capacity
(18)$$\begin{array}{ccc} h(p,T) & ≅ & h_{0}^{\prime }+c_{p0}^{\prime }\left(T-T_{0}\right)+\sum_{j}\frac{h\nu_{j}}{2}+\sum_{j}\frac{h\nu_{j}\;e^{-\beta h\nu_{j}}}{1-e^{-\beta h\nu_{j}}} \end{array}$$
(19)$$\begin{array}{ccc} s(p,T) & ≅ & s_{0}^{\prime }+c_{p0}^{\prime }\ln\left(T/T_{0}\right)+\frac{1}{T}\sum_{j}\frac{h\nu_{j}\;e^{-\beta h\nu_{j}}}{1-e^{-\beta h\nu_{j}}}\\ & - & k_{B}\sum_{j}\ln\left(1-e^{-\beta h\nu_{j}}\right) \end{array}$$
(20)$$\begin{array}{ccc} c_{p}\left(p,T\right) & ≅ & c_{p0}^{\prime }+\frac{1}{k_{B}T^{2}}\sum_{j}{\left[\frac{h\nu_{j}}{e^{\beta h\nu_{j}/2}-e^{-\beta h\nu_{j}/2}}\right]}^{2} \end{array}$$
where we used the harmonic approximation to express the vibrational partition function, i.e., $Q_{vb}≅\Pi_{j}\frac{e^{-\beta h\nu_{j}/2}}{1-e^{-\beta h\nu_{j}}}$, with $\nu_{j}$ being the quantum mode frequencies, which we always assume to be temperature independent. It is worth noting that the equations shown can be valid for temperatures where the ***q*** coordinates are well described as semi-classical degrees of freedom and, thus, Equations (Equation 18)–(Equation 20) cannot be used at low temperature conditions where pure quantum mechanical behavior is expected. We therefore consider only the $T\geq T_{0}$ temperature range, with $T_{0}$ being the lowest temperature still reasonably allowing us to treat the $q=\{q_{rt},q_{in}\}$ coordinates (the roto-translational coordinates $q_{rt}$ and the conformational coordinates $q_{in}$) as semi-classical degrees of freedom.

### 2.2. Parameterization Strategy

The Gamma state model described in the previous subsection requires knowledge of the quantum vibrational frequencies and the reference temperature excess enthalpy, entropy and heat capacity. While it is simple to evaluate its accuracy and obtain $c_{p0}^{\prime }$ by means of a linear fitting of the excess enthalpy change as provided by experiments or MD simulations over a large temperature range, the estimate of the quantum vibrational frequencies $\nu_{j}$, as well as of the reference temperature excess enthalpy and entropy ($h_{0}^{\prime },s_{0}^{\prime }$), requires a more complex procedure. In order to evaluate such parameters necessary to construct the isobaric equation of state, we identify a proper reference conformation (i.e., a free energy basin in conformational space) by means of an MD simulation at $T_{0}$ (hereafter. MD${}_{ref}$), where the $q_{in}$ coordinates can be treated as harmonic degrees of freedom. From the mass-weighted Hessian at the minimum energy structure of such a basin, we can obtain the frequencies of the quantum vibrational modes (we always assume that the quantum vibrational partition function is independent of the conformational coordinates). We can use the MD${}_{ref}$ trajectory to obtain the excess free energy, enthalpy and entropy differences ($\Delta \mu_{0}^{\prime },\Delta h_{0}^{\prime },\Delta s_{0}^{\prime }$) between the chosen conformation (the reference conformation) and the whole conformational space via
(21)$$\begin{array}{ccc} \Delta \mu_{0}^{\prime } & = & \mu_{0,ref}^{\prime }-\mu_{0}^{\prime }≅\mu_{ref}\left(T_{0}\right)-\mu \left(T_{0}\right)≅-k_{B}T_{0}\ln\left(P_{ref}\right) \end{array}$$
(22)$$\begin{array}{ccc} \Delta h_{0}^{\prime } & = & h_{0,ref}^{\prime }-h_{0}^{\prime }≅h_{ref}\left(T_{0}\right)-h\left(T_{0}\right)≅{\langle U_{pot}\rangle }_{ref,0}-{\langle U_{pot}\rangle }_{0} \end{array}$$
(23)$$\begin{array}{ccc} \Delta s_{0}^{\prime } & = & s_{0,ref}^{\prime }-s_{0}^{\prime }≅s_{ref}\left(T_{0}\right)-s\left(T_{0}\right)=\left(\Delta h_{0}^{\prime }-\Delta \mu_{0}^{\prime }\right)/T_{0}\\ & ≅ & \frac{{\langle U_{pot}\rangle }_{ref,0}-{\langle U_{pot}\rangle }_{0}}{T_{0}}+k_{B}\ln\left(P_{ref}\right) \end{array}$$
where the subscript ref indicates that the property is obtained within the reference conformation, $P_{ref}$ is the probability of the reference conformation as provided by the MD${}_{ref}$ trajectory, $U_{pot}$ is the potential energy due to the atomistic force field used in the MD simulations and ${\langle U_{pot}\rangle }_{ref,0},{\langle U_{pot}\rangle }_{0}$ are the corresponding average potential energies at $T_{0}$ within the reference conformation and over the whole conformational space, respectively. The ideal gas canonical partition function of the single *molecule* within the reference conformation at $T_{0}$, i.e., $Q_{ref}\left(T_{0}\right)$, can be written as
(24)$$\begin{array}{ccc} Q_{ref}\left(T_{0}\right) & ≅ & Q_{vb}\left(T_{0}\right)\;Q_{rt,ref}\left(T_{0}\right)\;Q_{in,ref}\left(T_{0}\right) \end{array}$$
(25)$$\begin{array}{ccc} Q_{rt,ref}\left(T_{0}\right) & ≅ & {\left(\frac{2\pi Mk_{B}T_{0}}{h^{2}}\right)}^{3/2}\;\frac{k_{B}T_{0}}{e^{-1}\;p}\;\frac{8\pi^{2}}{\left(1+\gamma \right)}\;\sqrt{I_{1}I_{2}I_{3}}\;{\left(\frac{2\pi k_{B}T_{0}}{h^{2}}\right)}^{3/2} \end{array}$$
(26)$$\begin{array}{ccc} Q_{in,ref}\left(T_{0}\right) & ≅ & e^{-\beta_{0}U_{e,ref}}\;\Pi_{j=1}^{n_{in}}\frac{e^{-\beta_{0}h\nu_{j,cl}/2}}{1-e^{-\beta_{0}h\nu_{j,cl}}} \end{array}$$
where $Q_{rt,ref}\left(T_{0}\right),Q_{in,ref}\left(T_{0}\right)$ are the roto-translational and semi-classical vibrational partition functions of the reference conformation, *M* is the *molecular* mass, 1+γ provides the quantum correction for the permutations of identical nuclei due to *molecular* rotations, $I_{1},I_{2},I_{3}$ are the moments of inertia as obtained at the reference conformation minimum energy structure and $U_{e,ref}$ and $\nu_{j,cl}$ are the corresponding electronic ground state energy and semi-classical mode frequencies (i.e., they are obtained at the reference conformation minimum energy structure). Moreover, $n_{in}$ is the total number of semi-classical modes and the reference conformation corresponds to the configurational subspace defined by considering for each semi-classical mode an interval of $\pm k\sigma_{j,cl}$ around the minimum energy position, with $\sigma_{j,cl}=\sqrt{k_{B}T}/\left(2\pi \nu_{j,cl}\right)$ and k>0 the largest integer number still providing the harmonic behavior within the reference conformation (i.e., a subspace allowing us to properly use Equation (Equation 26); see the next section for the criterion employed for choosing *k* in each system). Note that the factorization in Equation (Equation 24) follows from the (approximately) block diagonal *molecular* mass tensor uncoupling the roto-translational and internal momenta; in Equation (Equation 25). we disregard any degeneration or quasi-degeneration of the electronic ground state (e.g., due to nuclear spin states) and in Equation (Equation 26) we use the more general quantum harmonic expression for the semi-classical vibrational partition function instead of its classical limit, as $T_{0}$ is the boundary of the temperature range for treating $q_{in}$ as semi-classical degrees of freedom. From the last equations, we can obtain the *molecular* excess chemical potential, enthalpy and entropy at $T_{0}$ ($\mu_{0}^{\prime },h_{0}^{\prime },s_{0}^{\prime }$) via (see Equations (Equation 21)–(Equation 23)):
(27)$$\begin{array}{ccc} \mu_{0}^{\prime } & = & \mu_{0,ref}^{\prime }-\Delta \mu_{0}^{\prime }≅-k_{B}T_{0}\ln\left[q_{rt,ref}\left(T_{0}\right)\;q_{in,ref}\left(T_{0}\right)\right]\\ & + & k_{B}T_{0}+k_{B}T_{0}\ln\left(P_{ref}\right) \end{array}$$
(28)$$\begin{array}{ccc} h_{0}^{\prime } & = & h_{0,ref}^{\prime }-\Delta h_{0}^{\prime }≅U_{e,ref}+\sum_{j}\frac{h\nu_{j,cl}}{2}+\sum_{j}\frac{h\nu_{j,cl}\;e^{-\beta_{0}h\nu_{j,cl}}}{1-e^{-\beta_{0}h\nu_{j,cl}}}\\ & + & 4k_{B}T_{0}-\left({\langle U_{pot}\rangle }_{ref,0}-{\langle U_{pot}\rangle }_{0}\right) \end{array}$$
(29)$$\begin{array}{ccc} s_{0}^{\prime } & = & \frac{h_{0}^{\prime }-\mu_{0}^{\prime }}{T_{0}}≅\frac{U_{e,ref}}{T_{0}}+\frac{1}{T_{0}}\sum_{j}\frac{h\nu_{j,cl}}{2}+\frac{1}{T_{0}}\sum_{j}\frac{h\nu_{j,cl}\;e^{-\beta_{0}h\nu_{j,cl}}}{1-e^{-\beta_{0}h\nu_{j,cl}}}+3k_{B}\\ & - & \frac{1}{T_{0}}\left({\langle U_{pot}\rangle }_{ref,0}-{\langle U_{pot}\rangle }_{0}\right)+k_{B}\ln\left[\frac{q_{rt,ref}\left(T_{0}\right)\;q_{in,ref}\left(T_{0}\right)}{P_{ref}}\right] \end{array}$$
providing, via Equations (Equation 17)–(Equation 20), the complete *molecular* thermodynamics along the isobar. Note that in the results section, we will express the system enthalpy, entropy and chemical potential as the difference from the corresponding reference conformation property at T=0, $h_{ref}\left(0\right)$, $s_{ref}\left(0\right)$ and $\mu_{ref}\left(0\right)=h_{ref}\left(0\right)$, given by
(30)$$\begin{array}{ccc} h_{ref}\left(0\right) & = & h_{ref}^{\prime }\left(0\right)+\sum_{j}\frac{h\nu_{j}}{2} \end{array}$$
(31)$$\begin{array}{ccc} h_{ref}^{\prime }\left(0\right) & = & U_{e,ref}+\sum_{j}\frac{h\nu_{j,cl}}{2} \end{array}$$
(32)$$\begin{array}{ccc} s_{ref}\left(0\right) & = & 0 \end{array}$$
with $\nu_{j}$ the quantum mode frequencies.

Finally, it is worth remarking that we assume that within the whole temperature range considered, no mixing of quantum and semi-classical coordinates occurs, meaning that within each harmonic basin, the semi-classical modes obtained by the mass-weighted Hessian are always defined within the same configurational subspace, even if they can change from one basin to another. We actually estimate the number $n_{in}$ of the semi-classical internal coordinates (corresponding within the harmonic basin to the low-frequency modes), excluding from the total number of the internal coordinates all the stretching and bending degrees of freedom that we assume to be involved in the quantum modes (the high-frequency modes of the harmonic basin). Therefore, we identify the semi-classical vibrational modes within the reference conformation by the $n_{in}$ lowest-frequency modes (neglecting the roto-translational ones), assigning to the quantum modes all the other higher-frequency ones; i.e., we assume the dihedral angles and, if present, the libration coordinates as the internal semi-classical degrees of freedom.

## 3. Computational Details

The core of the computational part of the work is the production of reliable semi-classical simulations of the isolated species along an isobaric path. Note that, obviously, the isobaric condition is automatically fulfilled when a single *molecular* species is simulated resembling the ideal gas state. For this purpose, we utilized Gromacs software [44,45] version 5.1.2. The solutes, i.e., n-butane (hereafter termed as A), BmimNTf${}_{2}$ (hereafter termed as B), BmimPF${}_{6}$ (hereafter termed as C), n-butanol (hereafter termed as D), octadecane (hereafter termed as E) and n-octane (hereafter termed as F) were put at the center of an empty box of 125 nm${}^{3}$ volume. The temperature was kept constant using the Parrinello thermostat [46], the bond lengths were constrained using Lincs algorithm [47] and the electrostatics were taken into account using a cut-off of 0.8 nm and 1.1 nm for short-range and long-range electrostatics. All the simulations, carried out with a timestep of 2.0 fs from 250 K to 600 K, were extended up to 40 ns. The MD${}_{ref}$ simulation carried out at $T_{0}$ = 200 K was protracted, for all the five systems, by up to 100 ns to reduce the error possibly associated with the evaluation of the $P_{ref}$ and, hence, the excess free energy, enthalpy and entropy differences (Equations (Equation 21)–(Equation 23)) as described below. Once extracted from the MD${}_{ref}$ simulation (see next section), the reference structure, typically close to an accessible energy minimum, was further minimized, obtaining the reference conformation minimum energy structure according to the force field utilized for the simulation. Corresponding to this structure, the mass-weighted Hessian was then calculated, providing the associated harmonic frequencies excluding the rototranslations. The $n_{in}$ eigenvectors of the Hessian matrix corresponding to the $\nu_{j,cl}$ frequencies (i.e., the semi-classical modes determined as described in the Theory section and in the first part of the Results section) were then utilized to calculate $P_{ref}$. This was simply accomplished by considering the MD${}_{ref}$ simulation frames with projections on each of the $n_{in}$ semi-classical mass-weighted Hessian eigenvectors within $\pm k\sigma_{j,cl}$ around the minimum energy structure, with $\sigma_{j,cl}^{2}=k_{B}T_{0}/{\left(2\pi \nu_{j,cl}\right)}^{2}$ being the variance of the jth semi-classical mode coordinate. For each system studied, we determined *k* by comparing ${\langle U_{pot}\rangle }_{ref,0}-U_{pot,ref}$ with $\frac{N}{2}k_{B}T_{0}$, where $U_{pot,ref}$ is the MD force field potential energy of the reference minimum (typically, but not necessarily, the global potential energy minimum) and *N* is the total number of internal degrees of freedom of the simulated system (typically including the dihedral and the bending degrees of freedom). For each studied system, we chose the largest *k* such that, within the noise, ${\langle U_{pot}\rangle }_{ref,0}-U_{pot,ref}≅\frac{N}{2}k_{B}T_{0}$, thus ensuring the best statistical sampling and the accuracy of the harmonic approximation for the reference conformation at $T_{0}$. We actually considered the largest *k* values providing deviations between ${\langle U_{pot}\rangle }_{ref,0}-U_{pot,ref}$ and $\frac{N}{2}k_{B}T_{0}$ within either two (95 percent confidence) or three (99.9 percent confidence) standard errors of ${\langle U_{pot}\rangle }_{ref,0}$ (note that, once fulfilled, the criterion k≥3 guarantees that Equation (Equation 26) is a proper approximation). Note that we identified the reference conformation as the most sampled within the two-dimensional essential subspace as provided by the essential dynamics analysis of the $MD_{ref}$ trajectory, as described in detail elsewhere [48]. Such a choice should typically ensure that at low temperature (i.e., $T_{0}=200$ K), the reference minimum identified corresponds to the global minimum of the system. The interested reader can refer to Figure 1 herein reported for the steps described in the next section.

Obviously, the reliability of the present method in the form presented in this study, i.e., making use of a purely computational approach without any support from experimental data [30], entirely relies on the possible use of well-calibrated force fields capable of reasonably describing the system under investigation in the temperature range of interest (see above). For this reason, for all the six investigated systems, we utilized well-assessed force fields deposited in the Automatic Topology Builder REVISION 2021-05-20 [49,50]. Finally, on the same reference conformation minimum energy structure, we carried out quantum chemical calculations for obtaining the associated harmonic frequencies and moments of inertia necessary for properly evaluating the quantum vibrational, rototranslational and semi-classical vibrational canonical partition functions for the single *molecule*. These calculations were performed in the framework of the density functional theory using the wB97XD functional [51] in conjunction with the 6−31+G* basis set. The same level of theory was also adopted for further testing the quality of the selected force fields on a number of minimum energy configurations for the systems of interest. The Gaussian16-revision C.1 [52] program was employed for all the quantum chemical calculations. All the Cartesian coordinates of the optimized reference conformation geometries and the associated harmonic frequencies are reported in the Appendix A. A Fortran90 code to interface with the MD trajectory (xtc format), is available from the corresponding authors upon request.

## 4. Results and Discussion

After producing the MD simulations (both MD${}_{ref}$ and the simulations at the different temperatures (i.e., Steps 1 and 2 in the Figure 1), we evaluated the number of semi-classical modes corresponding to the $\nu_{j,cl}$ frequencies (the $n_{in}$ lowest frequency modes excluding the rototranslational ones) and assessed the accuracy of the diverging Gamma state. This was accomplished by a quadratic fitting of the simulation values of $\langle U_{pot,bending}\rangle$ and a linear fitting for $\langle U_{pot}\rangle -\langle U_{pot,bending}\rangle$, both as a function of temperature; $\langle U_{pot,bending}\rangle ≅U_{pot,bending}\left(0\right)+\frac{n_{b}}{2}k_{B}T+U_{pot,bending}^{{}^{\prime \prime }}\left(0\right)T^{2}/2$ is the MD average potential energy associated to the $n_{b}$ bending degrees of freedom (involved in the quantum modes) that we assume provide a quasi-harmonic contribution to the MD average potential energy with $U_{pot,bending}\left(0\right)$ and $U_{pot,bending}^{{}^{\prime \prime }}\left(0\right)$ the values of $\langle U_{pot,bending}\rangle$ and $\partial^{2}U_{pot,bending}/\partial T^{2}$ at T=0. $\langle U_{pot}\rangle$ is the MD average total potential energy and, thus, $\langle U_{pot}\rangle -\langle U_{pot,bending}\rangle$ is the MD average potential energy due only to the semi-classical degrees of freedom (in our simulations, all the stretching degrees of freedom were constrained). Note that, due to the high force constants, used to model the bending potential within the MD simulations, we can consider the simulated (classical) bending degrees of freedom as essentially uncoupled from the other degrees of freedom; thus, these latter coordinates are characterized by statistics that are virtually independent of the bending coordinates (i.e., identical to the statistics obtained constraining all the bending degrees of freedom).

The results shown in Figure 2 and Figure 3 clearly indicate that both $\langle U_{pot,bending}\rangle$ and $\langle U_{pot}\rangle -\langle U_{pot,bending}\rangle$ are remarkably linear in temperature, thus demonstrating the expected quasi-harmonic behavior of the bending coordinates and the accuracy of the diverging Gamma model for the semi-classical degrees of freedom. From the obtained $n_{b}$ and the slope of the linear fitting we then evaluated the number $n_{in}$ of the semi-classical modes, as well as the excess heat capacity $c_{p0}^{\prime }$. Note that the fitting parameter $U_{pot,bending}^{{}^{\prime \prime }}\left(0\right)$, in all cases almost negligible, is due to the slight anharmonicity of the (classical) bending degrees of freedom employed in the MD simulations, according to the MD force field used (in our statistical mechanical model, the bending and stretching contributions are always included via the harmonic quantum mode partition function $Q_{vb}$). It is also worth remarking that due to the data noise, the value of $n_{b}$ as obtained by the fitting of $\langle U_{pot,bending}\rangle$ may be not fully accurate, especially when dealing with systems involving a large number of bending degrees of freedom. Therefore, it is important when possible to check and correct the estimated number of semi-classical internal coordinates by comparing it with its direct evaluation, provided by summing the dihedral angles with, if present, the internal librational degrees of freedom.

Subsequently, as described in Methodology section and also in Figure 1, we evaluated the reference conformation probability at $T_{0}$
$P_{ref}$ and, hence, the excess free energy, enthalpy and entropy differences ($\Delta \mu_{0}^{\prime },\Delta h_{0}^{\prime },\Delta s_{0}^{\prime }$) between the chosen conformation (the reference conformation) and the whole conformational space. Finally, with the use of the reference minimum energy structure re-optimized at the DFT level (the corresponding coordinates are reported in the SI), we obtained the ideal gas canonical partition function of the reference conformation at $T_{0}$ and, thus, the required excess enthalpy and entropy at $T_{0}$ ($h_{0}^{\prime },s_{0}^{\prime }$) by means of Equations (Equation 27)–(Equation 29).

The obtained excess properties ($h_{0}^{\prime },s_{0}^{\prime },c_{p0}^{\prime }$), collected in Table 1, were finally utilized to obtain the complete thermodynamics (i.e., the isobaric equation of state) by means of Equations (Equation 17)–(Equation 20), the results of which are reported in Figure 4, Figure 5 and Figure 6.

In the case of n-butane (Figure 4A), our equation of state accurately reproduces the experimental enthalpy (with respect to $h_{ref}\left(0\right)$) at 300 K (18.3 kJ/mol experimental versus 17.9 kJ/mol calculated). The (standard-state) absolute entropy (302 mol${}^{-1}$ K${}^{-1}$ experimental [31] versus 326 J mol${}^{-1}$ K${}^{-1}$ calculated) at 272.7 K appears to be slightly overestimated, whereas the experimental isobaric heat capacity is accurately reproduced by our model in a wide temperature range (see Figure 6). Such results indicate that the n-butane MD force-field utilized (i.e., the dihedral potential) could provide an incorrect sampling at $T_{0}$ of the conformational space, overestimating the probability outside the reference conformation and resulting in a fixed and systematic entropy shift of about 24 J mol${}^{-1}$ K${}^{-1}$. Our results concerning the ion-pairs (Systems B and C) are in very good agreement with those provided by other computational methods reported in the literature [36,37]. In fact, we basically observed (see Figure 4) a superposition of our equation of state with the others, except for the entropy of BmimPF${}_{6}$, which is slightly larger in our equation of state (about 5 percent larger), possibly due to the anharmonic effects of the conformational space sampling that we explicitly include by means of the MD simulation data. The calculations reported by Kabo and coworkers [37], which are based on a mechanical sampling of each local minima described within the quasi-harmonic approximation, are likely to underestimate the configurational entropy due to the relative motions of the two partners of the ion-pair. For BmimNTf${}_{2}$ (Figure 4B), our calculations reproduce the experimental absolute entropy at 470 K almost exactly [34,36,53]. In the case of BmimPF${}_{6}$ (Figure 4C), the available experimental values for the entropy (reported in Figure 4 as filled circles [35,54]) appear to be less accurately reproduced by our calculations (relative deviations of ≈5–6 percent), similar to the results reported by Kabo and coworkers (see Figure 4). Interestingly the evaluated isobaric heat capacity is in good agreement with that calculated with different methods (see Figure 6B,C). It is important to note that the experimental values were derived by adding the measured standard evaporation entropy to the absolute entropy of the corresponding liquid phase, which was itself evaluated by integrating the experimental $c_{p}/T$ from 0 K (the so-called Third Law method). The evaporation entropy is typically obtained from the y-axis intercept of the extrapolated Clausius–Clapeyron fitting line in a lnp vs 1/T graph. The thus-obtained value is assigned to the mean temperature of the experiments. Note that, besides the uncertainty in the evaporation entropy, the determined gas entropy values suffer from possible inaccuracies in the entropy of the liquid phase. For example, discrepancies exceeding 10 percent were reported in the literature for the experimental heat capacities of liquid BMImPF_6_ [56]. Moreover, if the experimental entropy of the liquid is available at temperatures lower than those explored in evaporation experiments, as can be the case, an extrapolation to the mean temperature of the evaporation measurements must be performed. It should also be noted that the co-occurrence of thermal decomposition processes during evaporation was reported for a number of ionic liquids, including BmimPF${}_{6}$ [35], which can seriously affect the measured mass loss and vapor pressures. From Figure 5 and Figure 6D–F, it is evident that our results for n-butanol (D), octadecane (E) and n-octane (F) are accurate in reproducing the available experimental data (an are also in agreement with the results from other theoretical-computational methods), confirming the reliability of the proposed theoretical-computational approach and suggesting a higher accuracy for the typical MD force field, even at low temperatures, as the system complexity increases (i.e., a larger number of internal semi-classical degrees of freedom).

## 5. Concluding Remarks

In this study, we have presented a theoretical-computational procedure for calculating the thermodynamic properties of flexible gaseous molecular systems as a function of temperature. The obtained analytical (isobaric) equation of state, providing the explicit temperature dependence of all the relevant thermodynamic properties, proved to rather accurately reproduce the experimental thermodynamics of six molecular and supramolecular systems of different complexity. In particular, we tested the method on the following systems: (i) n-butane, an extensively investigated system both experimentally and computationally; (ii) the ion pairs known to mainly represent the vapor-phase in equilibrium over the ionic liquids 1-butyl-3-methylimidazolium hexafluorophosphate (BmimPF${}_{6}$) and 1-butyl-3-methylimidazolium bis(trifluoromethylsulfonyl)imide (BmimNTf${}_{2}$); (iii) n-butanol; (iv) octadecane; and (v) n-octane. The proposed method is entirely based on the application on the QGE theory using, as input data, the results of MD simulations. For this reason, its reliability strongly depends on the physical consistency of the semi-classical atomistic simulations, in particular, the quality of the adopted force field and the lack of relevant electronic transitions (e.g., intramolecular or intra-complex charge transfer or chemical reactions) accompanying the molecular conformational changes. Moreover, the application of the method is subject to the possibility of separating, in a non-arbitrary way, semi-classical and quantum internal modes. The proposed method allows us to obtain the complete thermodynamics of the molecular system of interest over a temperature range whose extent must ensure the consistency of the force field and the MD simulations. If compared to other methods proposed in the past for the same purpose, our approach has the advantage of being specifically suited for complex molecular–supramolecular systems (i.e., involving several internal semi-classical degrees of freedom), for which conformational sampling may represent a serious computational bottleneck, not only in terms of computational cost but also in terms of the definition of the actual conformationally relevant coordinates.

## Figures and Tables

**Figure 1 molecules-27-07863-f001:**
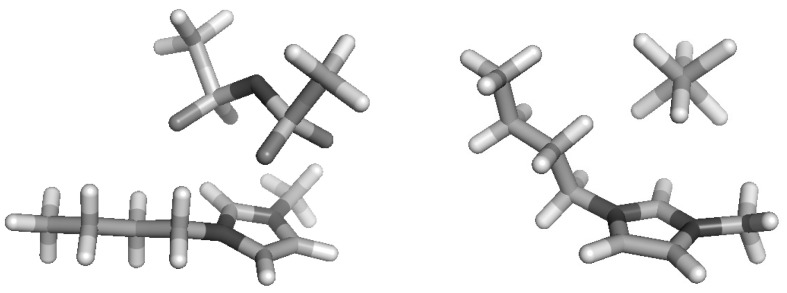
Pictorial view of the structures BmimNTf${}_{2}$ and BmimPF${}_{6}$.

**Scheme 1 molecules-27-07863-sch001:**
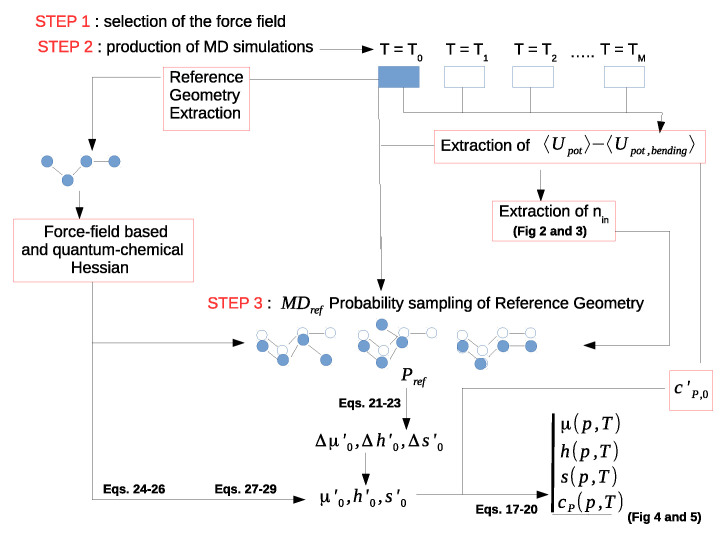
Scheme I.

**Figure 2 molecules-27-07863-f002:**
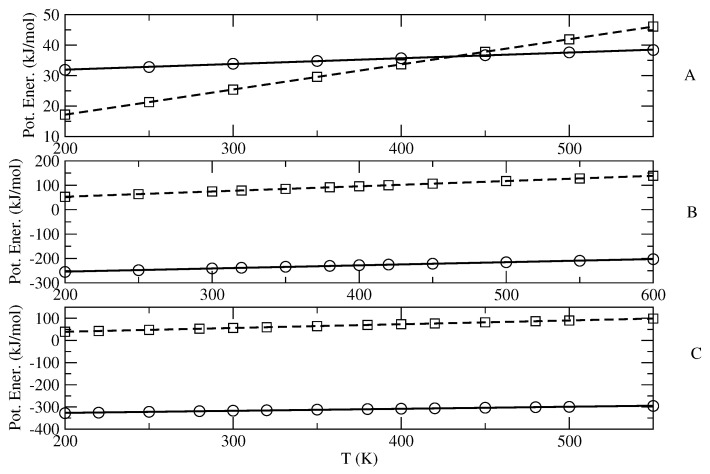
Plot of $\langle U_{pot,bending}\rangle$ (squares) and $\langle U_{pot}\rangle$ − $\langle U_{pot,bending}\rangle$ (circles) as a function of the temperature, provided by the MD simulations of n-butane (**A**), BmimNTf${}_{2}$ (**B**) and BmimPF${}_{6}$ (**C**), as well as their quadratic and linear fittings (dashed and solid lines, respectively).

**Figure 3 molecules-27-07863-f003:**
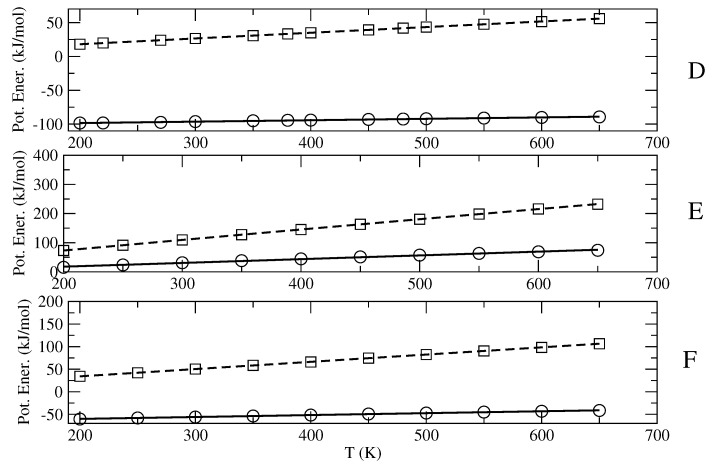
Plot of $\langle U_{pot,bending}\rangle$ (squares) and $\langle U_{pot}\rangle$ − $\langle U_{pot,bending}\rangle$ (circles) as a function of the temperature, provided by the MD simulations of n-butanol (**D**), octadecane (**E**) and n-octane (**F**), as well as their quadratic and linear fittings (dashed and solid lines, respectively).

**Figure 4 molecules-27-07863-f004:**
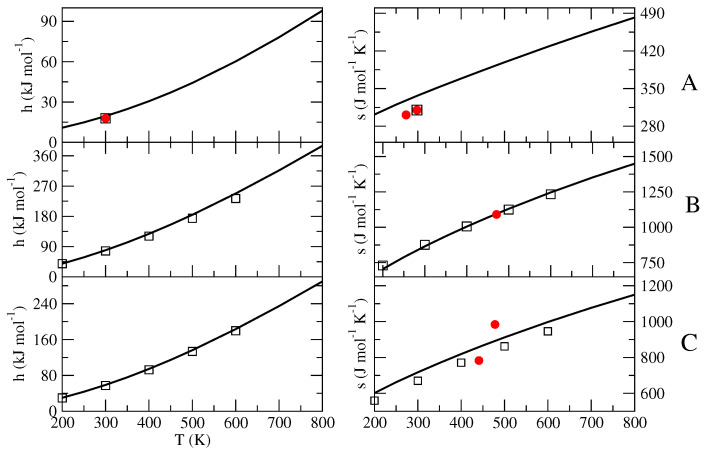
Standard-state molecular full enthalpy (with respect to $h_{ref}\left(0\right)$) and entropy as a function of the temperature for n-butane (**A**), BmimNTf${}_{2}$ (**B**) and BmimPF${}_{6}$ (**C**), as provided by our equation of state (solid line). The available experimental data found in the literature are reported with filled red circles: for (**A**) from references [31,32]; for (**B**) from references [34,36,53]; for (**C**) from references [35,54]. Values calculated with different theoretical-computational procedures are reported with squares: for (**A**) from reference [17]; for (**B**) from reference [36]; for (**C**) from reference [37].

**Figure 5 molecules-27-07863-f005:**
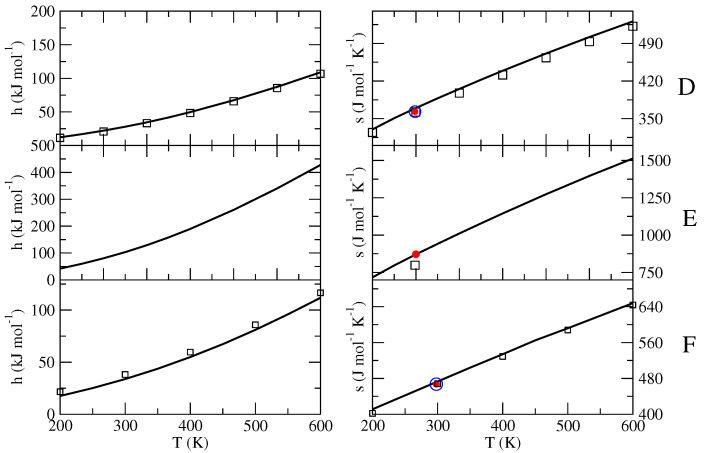
Standard-state molecular full enthalpy (with respect to $h_{ref}\left(0\right)$) and entropy as a function of the temperature for n-butanol (**D**), octadecane (**E**) and n-octane (**F**), as provided by our equation of state (solid line). The available experimental data found in the literature are reported with filled red circles: for (**D**) from reference [39]; for (**E**) from reference [5]; for (**F**) from references [40,41]. Values calculated with different theoretical-computational procedures are reported with squares and blue circles: for (**D**) from reference [38] (squares) and from reference [20] (blue circle); for (**F**) from reference [55] (squares) and from reference [5] (blue circle).

**Figure 6 molecules-27-07863-f006:**
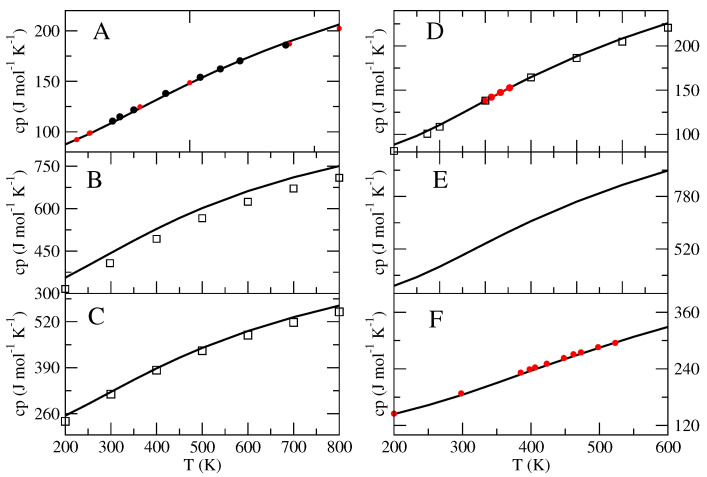
Comparison of the molecular isobaric heat capacity for n-butane (**A**), BmimNTf${}_{2}$ (**B**), BmimPF${}_{6}$ (**C**), n-butanol (**D**), octadecane (**E**) and n-octane (**F**), as provided by our equation of state (solid line). Experimental values are reported with filled red or black circles: for (**A**) from reference [31,32]; for (**D**) from reference [39]; for (**F**) from reference [40,41]. Values calculated with different theoretical-computational procedures are reported with squares: for (**B**) from reference [36]; for (**C**) from reference [37]; for (**D**) from reference [38].

**Table 1 molecules-27-07863-t001:** Excess molecular free energy, enthalpy and entropy differences as obtained from $MD_{ref}$ at $T_{0}=200$ K using the largest *k* value compatible, within either 95 percent (left numbers) or the 99.9 percent (right numbers) confidence, with the harmonic assumption for the reference conformation (see Computational Details section), with the corresponding standard-state molecular excess enthalpy (with respect to $h_{ref}^{\prime }\left(0\right)$) and entropy at $T_{0}$, as obtained from Equations (Equation 27)–(Equation 29), and the molecular excess heat capacity, as provided by the MD simulations at the different temperatures ($n_{in}$ indicates the number of semi-classical internal degrees of freedom). A = n-butane, B = BmimNTf${}_{2}$, C = BmimPF${}_{6}$, D = n-butanol, E = octadecane and F = n-octane. Note that for E and F, a single value of *k* is reported, corresponding to 95 percent confidence; for any larger *k*, we could not find conditions within 99.9 percent confidence (i.e., deviations between the MD mean potential energy and the expected harmonic value are too large).

	$n_{\mathrm{in}}$	*k*	$\Delta \mu_{0}^{\prime }$	$\Delta h_{0}^{\prime }$	$\Delta s_{0}^{\prime }$	$h_{0}^{\prime }-h_{\mathrm{ref}}^{\prime }\left(0\right)$	$s_{0}^{\prime }$	$c_{p0}^{\prime }$
			kJ/mol	kJ/mol	J/(mol K)	kJ/mol	J/(mol K)	J/(mol K)
A	3	7	5.4	−0.65	−30.3	9.9	302.1	64.5
		20	4.1	−0.51	−23.1	9.7	295.4	64.5
B	23	7	10.1	−0.9	−55.1	33.4	669.7	257.8
		8	9.1	−0.53	−48.0	33.0	662.7	257.8
C	18	6	13.4	−1.2	−73.1	26.2	591.1	200.2
		8	10.9	−2.3	−66.1	27.3	584.1	200.2
D	4	4	8.2	−1.0	−46.5	11.0	334.3	70.7
		8	6.0	−1.1	−35.0	11.1	322.8	70.7
E	17	4	12.6	−6.2	−93.7	33.8	654.7	233.1
F	7	7	7.1	−0.4	−37.4	14.7	393.7	104.8

## Data Availability

Not applicable.

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
