# Peer review of "Theoretical-Computational Modeling of Gas-State Thermodynamics in Flexible Molecular Systems: Ionic Liquids in the Gas Phase as a Case Study"

_molecules, 2022, doi:10.3390/molecules27227863_

Round 1

Reviewer 1 Report

Comments:

I have completed the evaluation of the paper “Theoretical-computational modeling of gas-state thermodynamics in flexible molecular systems: ionic liquids in the gas-phase as a case study.” by Andrea Amadei et al. submitted to Molecules for its possible publication. The authors propose A theoretical-computational procedure based on the quasi-Gaussian-Entropy (QGE) theory and Molecular Dynamics (MD) simulations for the calculation of the thermodynamic properties for molecular and supramolecular species in the gas-phase. The method was tested on six model systems consisting of conventional solvents and ionic liquids, and showed the reasonable predicted results. Thus, it can be published after addressing the following comments:

(1) Lines 101-110: The subscript of the chemical formulas e.g., C4H10O, BmimNTf2, and C18H38 should be modified.

(2) The section “Results and discussion” should be added contents for more detailed discussion.

(3) The experimental data should be added into the figures to further validate the reliability of the proposed theoretical method.

(4) All of the references should be carefully checked because of the inconsistent style.

For example, the journal name in some references was not be abbreviated.

41. Berendsen, H.J.C.; van der Spoel, D.; van Druned, R. GROMACS: A message-passing parallel molecular dynamics implementation, Computer Physics Communications 1995, 91, 43-56.

46. Malde A.K.; Zuo L.; Breeze M.; Stroet M.; Poger D.; Nair P.C.; Oostenbrink C.; Mark A.E. An Automated force field Topology Builder (ATB) and repository: version 1.0. J. Chem. Theory and Comp., 2011, 7, 4026-4037.

52. Zaitsau, D.H.; Yermalayeu, A.V.; Emel’yanenko, V.N.; Butler, S.; Schubert, T.; Verevkin, S.P. Thermodynamics of Imidazolium-Based Ionic Liquids Containing PF6 Anions. J. Phys. Chem B 2016 120, 7949-7957

Author Response

First of all we wish to thank the referee for his/her comments. Below we report the responses to the specific points raised by the referee.

(1) Lines 101-110: The subscript of the chemical formulas e.g., C4H10O, BmimNTf2, and C18H38 should be modified.

This has been done as requested by the referee

(2) The section “Results and discussion” should be added contents for more detailed discussion.

We are willing to include more contents, but we didn’t fully understand what exactly the referee is referring to. If the referee could explain us which part of the discussion should be extended, we’ll do it.

(3) The experimental data should be added into the figures to further validate the reliability of the proposed theoretical method.

Actually all the available experimental data (found in the literature) have been included in the Figures as explicitly mentioned in the Figures captions.

(4) All of the references should be carefully checked because of the inconsistent style.

This has been done, as requested by the referee.

Reviewer 2 Report

The present manuscript showcases a computational procedure for calculating the thermodynamic properties of flexible gas phase systems.  It essentially produces an equation of state using as input data MD simulations from which relevant thermodynamic properties can be extracted. The procedure is tested by reproducing experimental thermodynamics of six molecules of systems increasing complexity. 

The theoretical development is sound, and actually is a continuation of previous papers by some of the authors. In this manuscript, the application of the model is presented, hence serves as an appropriate template for researcher wishing to employ the methods.

The excelling feature of the method is that the sampling of conformational space is improved as compared to traditional approaches, hence there is a possible advantage when applied to large and complex molecules.

The paper is well written, although a proofreading of the manuscript is in order,  ( e.g. foundamental interest ; gasesous molecular , etc.)

Otherwise, the manuscript is a solid contribution and should be considered for publication.

Author Response

We thank the referee for her/his positive comments.

We have checked for the possible typos and corrected them when necessary.

Round 2

Reviewer 1 Report

This manuscript can be considered publication in the current version.